# The Role of Adaptive Optimizers
# for Honest Private Hyperparameter Selection

## Abstract

Hyperparameter optimization is a ubiquitous challenge in machine learning, and the performance of a trained model depends crucially upon their effective selection. While a rich set of tools exist for this purpose, there are currently no practical hyperparameter selection methods under the constraint of differential privacy (DP). We study honest hyperparameter selection for differentially private machine learning, in which the process of hyperparameter tuning is accounted for in the overall privacy budget. To this end, we i) show that standard composition tools outperform more advanced techniques in many settings, ii) empirically and theoretically demonstrate an intrinsic connection between the learning rate and clipping norm hyperparameters, iii) show that adaptive optimizers like DPAdam enjoy a significant advantage in the process of honest hyperparameter tuning, and iv) draw upon novel limiting behaviour of Adam in the DP setting to design a new and more efficient optimizer.

## 1  Introduction

Over the last several decades, the field of machine learning has flourished. However, training machine learning models frequently involves personal data, which leaves data contributors susceptible to privacy attacks. This isn't purely hypothetical: recent results have shown that models are vulnerable to membership inference [SSSS17, CLE$^+$19, NSH19] and model inversion attacks [FJR15, SRS17]. The leading approaches for privacy-preserving machine learning are based on differential privacy (DP) [DMNS06]. Informally, DP rigorously limits and masks the contribution that an individual datapoint can have on an algorithm's output. To address the aforementioned issues, DP training procedures have been developed [WM10, BST14, SCS13, ACG$^+$16], which generally resemble non-private gradient-based methods, but with the incorporation of gradient clipping and noise injection.

In both the private and non-private settings, *hyperparameter selection* is instrumental to achieving high accuracy. The most common methods are grid search or random search, both of which incur a computational overhead scaling with the number of hyperparameters under consideration. In the private setting, this issue is often magnified as most private training procedures introduce new hyperparameters. Regardless, and more importantly, hyperparameter tuning on a sensitive dataset also costs in terms of *privacy*, naively incurring a multiplicative cost which scales as the square root of the number of candidates (based on composition properties of differential privacy [KOV15]).

Most prior works on private learning choose not to account for this cost [ACG$^+$16, YLP$^+$19, TB21], focusing instead on demonstrating the accuracy achievable by private learning under idealized conditions, that is, if the best hyperparameters were somehow known ahead of time. Some works assume the presence of supplementary public data resembling the sensitive dataset [AGD$^+$20, RTM$^+$20], which may be freely used for hyperparameter tuning. Naturally, such public data may be scarce or nonexistent in settings where privacy is a concern, leaving practitioners with little guidance on how

to choose hyperparameters in practice. As explored in our paper, poor hyperparameter selection with standard private optimizers can have catastrophic effects on model accuracy.

Hope is afforded by the success of adaptive optimizers in the non-private setting. The canonical example is Adam [KB14], which exploits moments of the gradients to adaptively and dynamically determine the learning rate. It works out of the box in many cases, providing accuracy comparable with tuned SGD. We navigate the different options available to a practitioner to solve the *honest private hyperparameter tuning problem* and ask, *are there optimizers which provide strong privacy, require minimal hyperparameter tuning, and perform competitively with tuned counterparts?*

**Our Contributions**

1. We investigate techniques for private tuning of hyperparameters. We perform the first empirical evaluation of the proposed theoretical method of Liu and Talwar [LT19], and demonstrate that it can be relatively expensive. That is, in certain cases, one can tune over sufficiently many hyperparameters using standard composition tools such as moments accountant [ACG$^+$16].

2. We empirically and theoretically demonstrate that two hyperparameters, the learning rate and clipping threshold, are intrinsically coupled for non-adaptive optimizers. While other hyperparameters and the model architecture are restricted by the scope of the task, privacy and utility targets, and computational resources, the learning rate and clipping norm have no a priori bounds. Since the resulting hyperparameter grid adds up to the privacy cost while tuning to achieve the model with the best utility, we explore leveraging adaptive optimizers to reduce the hyperparameter space.

3. We empirically demonstrate the DPAdam optimizer (with default values for most hyperparameters), can match the performance of tuned non-adaptive optimizers on a variety of datasets, thus enabling private learning with honest hyperparameter selection. This finding complements a prior claim of Papernot et al. [PCS$^+$20], which suggests that a well-tuned DPSGD can outperform DPAdam. However, our findings show that this difference in performance is relatively insignificant. Furthermore, in the realistic setting where hyperparameter tuning must be accounted for in the privacy loss, we show that DPAdam is much more likely to produce non-catastrophic results.

4. We show that the adaptive learning rate of DPAdam converges to a static value. To leverage this, we introduce a new private optimizer, DPAdamWOSM that matches the performance of DPAdam without computing the second moments.

## 1.1 Related Work

Hyperparameter tuning plays a vital role in machine learning practice. In the non-private setting, ML practitioners use grid search, random search, Bayesian optimization techniques [SSA13] or AutoML [HZC21] techniques to tune their models. However, there hasn't been much research on private hyperparameter tuning procedures due to the significant associated privacy costs. Each set of hyperparameter configuration results in a privacy-utility tradeoff. This tradeoff for multiple configurations can be captured by Pareto frontiers using multivariate Bayesian optimization over parameter dimensions [AGD$^+$20]. However, this method asks the model curator to query the dataset multiple times which requires non-private access to the dataset. There have been some end-to-end private tuning procedures [CMS11, CV13, KGGW15] which work for a selected number of hyperparameter sets. These results work either in restricted settings for few combinations of candidates or under relaxations of differential privacy. The most relevant work to ours is an approach for private selection from private candidates [LT19]. Their work provides two methods, one which outputs a candidate with accuracy greater than a given input threshold, and another which randomly stops and outputs the best candidate seen so far. The first approach is of limited utility in practice as it requires a prior accuracy bound for the dataset. The second variant incurs a considerable overhead in the privacy cost. We study this second approach and compare it with naive approaches based on Moments Accountant [ACG$^+$16] which would scale as the square root of the number of candidates.

## 2 Problem Setup and Overview

Consider a sensitive dataset $D$ which lies beyond a privacy firewall and has $n$ points of the form $(x_1, y_1), (x_2, y_2), \ldots, (x_n, y_n)$ where $x_i \in \mathcal{X}$ is the feature vector of the $i$th point and $y_i \in \mathcal{Y}$ is its desired output. Though our experiments are carried out in the supervised setting, all results can be

translated to unsupervised setting as well. The dataset has been divided into two parts, the training set and the validation set. A trusted curator wants to train a machine learning model by making queries on the dataset with a total end-to-end training privacy budget of $(\varepsilon_f, \delta_f)$ such that the model can perform with high accuracy on the validation set. The curator wants to try multiple hyperparameter candidates for the model to figure out which candidate gives the maximum accuracy. However, as the model is private, each candidate requires multiple queries made on the dataset and all of them need to be accounted in the total privacy budget of $(\varepsilon_f, \delta_f)$.

Note that any validation accuracy must also be measured privately. Since this accuracy is a low-sensitivity query with a scalar output, and must only be computed once per choice of hyperparameters, the cost of this procedure is generally a lower order term versus the main training procedure. Thus for simplicity, we do not noise these validation accuracy queries. As we will see later, some optimizers require more candidates to tune and hence would also require more privacy budget than others.

To tackle private hyperparameter selection, we first compare the available private tuning procedures in Section 3. We show that the privacy cost for training a model depends on the hyperparameter grid size and standard composition theorems provide the best guarantees when the grid is small. In Section 4, we investigate different optimizers to see how many candidates are required to output a good solution. In Section 4.1 we provide theoretical and empirical evidence to demonstrate an intrinsic coupling between two hyperparameters – the learning rate and clipping norm in DPSGD. We show that this coupling makes DPSGD sensitive to these parameter choices, which can drastically affect the validation accuracy. In Section 4.2 we demonstrate that an adaptive optimizer, DPAdam, translates well from the non-private setting and obviates tuning of the learning rate. In Section 5, we empirically compare DPAdam with DPSGD and DPMomentum to show that DPAdam performs at par with less hyperparameter tuning. Finally, in Section 6, we establish that DPAdam converges to a static learning rate in restricted settings, and unveil a new optimizer DPAdamWOSM which can leverage this converged value without computing the second moments. In the interest of space, we defer standard preliminaries such as DP definitions, hyperparameters, and optimizers to the appendix.

## 3 The Cost of Privately Tuning DP Optimizers

Effective hyperparameter tuning is crucial in extracting good utility from an optimizer. Unlike the non-private setting, DP optimizers typically i) have more hyperparameters to tune; ii) require additional privacy budget for tuning. Existing work on DP optimizers acknowledge this problem (e.g., [ACG+16]), but do not address the privacy cost incurred during hyperparameter tuning [ACG+16, YLP+19, TB21]. There are two main prior general-purpose approaches for private hyperparameter selection. The first performs composition via Moments Accountant [ACG+16], and the second is the algorithm of Liu and Talwar (LT) [LT19]. The latter is a theoretical result, and to the best of our knowledge, has not been previously evaluated in practice. We investigate the privacy cost of these two techniques in practice and discuss situations in which each method is preferred.

### 3.1 Hyperparameter Selection via [LT19]

Liu and Talwar [LT19] propose a random stopping algorithm (LT) to output a 'good' hyperparameter candidate from a pool of $K$ candidates, $\{x_1, \ldots, x_K\}$. They assume sampling access to a randomized mechanism $Q(D)$ which samples $i \sim [K]$, and returns the $i$-th candidate $x_i$, and a score $q_i$ for this candidate. It is a random stopping algorithm, in which at every iteration, a candidate is picked from $Q$ i.i.d. with replacement and a $\gamma$-biased coin is flipped to randomly stop the algorithm. When the algorithm stops, the candidate with the maximum score seen so far is outputted. In the approximate DP version of this algorithm, an extra parameter $\Upsilon$ is set to limit the total of number of iterations. The pseudocode of this algorithm is deferred to the appendix.

**Theorem 1** ([LT19], Theorem 3.4)**.** *Fix any $\gamma \in [0, 1]$, $\delta_2 > 0$ and let $\Upsilon = \frac{1}{\gamma} \log \frac{1}{\delta_2}$. If $Q$ is $(\varepsilon_1, \delta_1)$-DP, then the LT algorithm is $(\varepsilon_f, \delta_f)$-DP for $\varepsilon_f = 3\varepsilon_1 + 3\sqrt{2\delta_1}$ and $\delta_f = \sqrt{2\delta_1}\Upsilon + \delta_2$.*

Theorem 1 expresses the privacy cost of the algorithm in terms of the privacy cost of individual learners, and parameters of the algorithm itself. The $\delta_2$ parameter does not significantly affect the final epsilon $\varepsilon_f$ of the algorithm and in practice, one can set it to a very small value ($10^{-20}$). Though a small value of $\delta_2$ has little effect on $\delta_f$, it increases the hard stopping time of the algorithm, $\Upsilon$.

To understand the LT algorithm, we will compare the privacy costs of training a single hyperparameter candidate with a final $\varepsilon_f, \delta_f$ budget via LT and compare it with the privacy cost $\varepsilon_1, \delta_1$ of the underlying individual learner. This setting might seem unnatural for LT as it was designed to select from a pool of candidates but we choose this setting to show the minimum privacy cost overhead associated with LT and later show how the privacy cost changes for multiple candidates (varying $\gamma$). To use LT, one needs to figure out the $\varepsilon_1, \delta_1$ via Theorem 1 using the final $\varepsilon_f, \delta_f$ values and in this case, $\gamma = 1$ (as we have just one candidate). The individual learner is then trained using $\varepsilon_1, \delta_1$ budget.

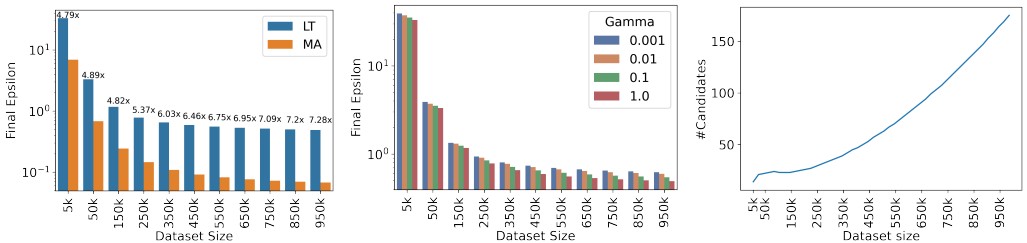

Figure 1: Comparing the privacy cost of LT versus Moments Accountant. The minimal privacy overhead incurred by LT is at least ~5x, and increases with the dataset size (left). However, as we allow LT to sample and test more candidate hyperparameters, the privacy cost barely increases (middle). Moments Accountant is able to test a significant number of candidates at the same cost as the minimal privacy overhead of LT (right).

Due to the delicate balance of $\delta_f$ in Theorem 1, one can see the $\delta_1$ comes out to be much smaller than $\delta_f$. This change in $\delta_1$ results in a blowup of $\varepsilon_1$ and hence, the final privacy cost of the LT algorithm $(3\varepsilon_1 + 3\sqrt{2\delta_1})$, is much larger than what it would have been for learning one candidate without LT. We call this increase the *blowup* of privacy. We measure this blowup in Figure 1(left), for the setting of $\sigma = 4, L = 250, T = 10,000$ with varying dataset sizes ($n$). It can be seen that for $n = 5,000$, the blowup is 4.8x whereas for for $n = 950,000$, the blowup is almost 7.3x (note the log scale on y-axis). Qualitatively similar trends persist for other choices of noise multiplier, lot size and iterations. We add more experiments to compare LT vs MA with varying candidate sizes in the appendix.

Furthermore, we show that although LT entails a privacy blowup, decreasing $\gamma$, which corresponds to training more individual learners with $\varepsilon_1, \delta_1$, doesn't result in a significant difference in the final epsilon guaranteed by LT. In Figure 1(middle), we show the final epsilon cost for different dataset size and varying values of $\gamma \in [0.001, 0.01, 0.1, 1]$. It is interesting to note here that with smaller $\gamma$ values, one can train many candidates (in expectation, $\frac{1}{\gamma}$) for negligible additional privacy cost. The blowup to train 1 candidate ($\gamma = 1$) versus $1,000$ candidates ($\gamma = 0.001$) increases from 33 to 39 for $n = 5,000$ and increases from 0.49 to 0.69, for $n = 950,000$. This increase is minimal in comparison to advanced composition, which grows proportional to $\mathcal{O}(\sqrt{k})$. However, another resource at play is the total training time, which is proportional to $1/\gamma$ (i.e., the total number of candidates). In summary, the LT algorithm is effective if an analyst has the privacy budget to afford the initial blowup, as the privacy cost of testing additional hyperparameters is insignificant.

## 3.2 Hyperparameter Selection via Moments Accountant

We learnt from the previous section that, LT permits selection from a large pool of hyperparameters (depending on the $\gamma$ value) but incurs a constant privacy blowup. We compare LT with tuning using Moments Accountant (MA). We notice using that with the same initial privacy blowup of the LT algorithm, MA is able to compose a considerable number of hyperparameter candidates. In Figure 1(right), we show the number of candidates that can be composed using MA for the minimum privacy cost for running the LT algorithm ($\gamma = 1$), for the setting of $\sigma = 4, L = 250, T = 10,000$ and varying dataset size ($n$) on the x-axis. As the $T$ and $L$ is set constant, bigger $n$ values in this graph correspond to fewer epochs of training and hence, worse utility. Depending on dataset size, MA can compose 14 candidates for $n = 5000$ and up to 175 candidates when $n = 100000$. It is perhaps surprising how well a standard composition technique performs versus LT. This information can be highly valuable to a practitioner who has limited privacy budget. Qualitatively similar trends persist for other choices of batch size, noise multiplier, and iterations.

From our experiments for both these tuning procedures, we conclude that while tuning with LT entails an initial privacy blowup, and the additional privacy cost for trying more candidates (smaller $\gamma$) is minimal. Even though this has an additional computation cost, it can be appealing when an analyst wants to try numerous hyperparameters. On the other hand, for the same overall privacy cost, MA can be used to compose a significant number of hyperparameter candidates. Additionally, MA allows access to all intermediate learners, whereas LT allows access to only the final output parameters. In the sequel, this conclusion will be useful in making the naive MA approach a more appealing tool for some settings (e.g., tighter privacy budgets).

## 4 Tuning DP Optimizers

We detail aspects of tuning both non-adaptive and adaptive optimizers. We start with tuning non-adaptive optimizers (Section 4.1). We theoretically and empirically demonstrate a connection between the learning rate and clipping threshold. We also establish that non-adaptive optimizers inevitably require searching over a large LR-clip grid to extract performant models. Adaptive optimizers forego this problem as they do not need to tune the hyperparameter dimension of learning rate. However, they introduce other hyperparameters that have known good choices in the non-private setting, and we empirically show that they are good candidates in the private setting (Section 4.2).

### 4.1 Tuning DP non-adaptive optimizers

While many hyperparameters are restricted due to computational and privacy/utility targets, the learning rate $\alpha$ and the clipping threshold $C$ have no a priori bounds. In what follows, we show an interplay between these parameters by first theoretically analyzing the convergence of DPSGD. We then explore an illustrative experiment which demonstrates their entanglement. In the following theorem, we derive a bound on the expected excess risk of DPSGD and while doing so, show that the optimal learning rate, $\alpha_{opt}$, is proportional to the inverse of $C$. The proof appears in Appendix D.

**Theorem 2.** *Let $f$ be a convex and $\beta$-smooth function, and let $x^* = \underset{x \in \mathcal{S}}{\arg \min} f(x)$. Let $x_0$ be an arbitrary point in $\mathcal{S}$, and $x_{t+1} = \Pi_{\mathcal{S}}(x_t - \alpha(g_t + z_t))$, where $g_t = \min(1, \frac{C}{\|\nabla f(x)\|^2})\nabla f(x)$ and $z_t \sim \mathcal{N}(0, \sigma^2 C^2)$ is the noise due to privacy. After $T$ iterations, the optimal learning rate is $\alpha_{opt} = \frac{R}{CT\sqrt{1+\sigma^2}}$, where $\mathbb{E}[f(\frac{1}{T}\sum_i^T x_t) - f(x^*)] \leq \frac{RC\sqrt{1+\sigma^2}}{\sqrt{T}}$ and $R = \mathbb{E}[\|x_0 - x^*\|]$.*

Though Theorem 2 gives a closed-form expression for the optimum learning rate, it is a function of the parameter $R$, which is unknown a priori to the analyst. Given constant $T$ and $\sigma$, the optimal learning rate $\alpha_{opt}$ is inversely proportional to the clipping norm $C$. This is crucial information in practice because these parameters vary among datasets and are unbounded. This unboundedness property thus requires us to search over very large ranges of $C$ and $\alpha$ when we have no prior knowledge of the dataset. It is natural to ask whether one can fix the clipping norm $C$ and search only over a wide range for the learning rate $\alpha$ (or vice versa). We explore this relationship experimentally, showing that fixing one of these two hyperparameters may often but not always result in an optimal model.

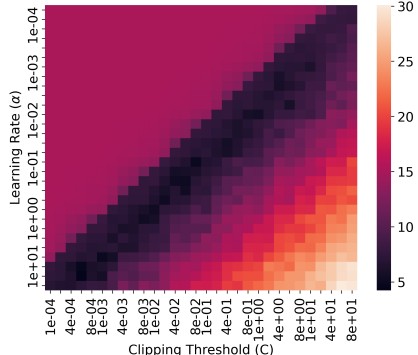

Figure 2: Log of training loss for simulation experiment at $\sigma = 4$ on a synthetic dataset. The black pixels correspond to lowest training loss. Note that most best loss values lie on a diagonal expressing the inverse connection between $\alpha$ and $C$.

In this experiment, we train a linear regression model on a 10-dimensional synthetic dataset of input-label pairs $(x, y)$ sampled from a distribution $\mathcal{D}$ as follows: $x_1, \ldots, x_d \sim \mathcal{U}(0, 1), y = x \cdot w^*, w^* = 10 \cdot \mathbf{1}^d$. We use the initialization $w_0 = \mathbf{0}^d$ and train for 100 iterations. In the non-private setting, this model converges quickly with any reasonable learning rate, but in the private setting, we notice that the training loss depends heavily on the choice of $\alpha$ and $C$. Figure 2 shows a heat map for the log training loss when trained on $(\alpha, C)$ pairs taken from a large grid consisting of $[1, 2, 4, 5, 8]$ at scales of $[10^{-4}, 10^{-3}, 10^{-2}, 10^{-1}, 10^0, 10^1]$. The best training is observed when the loss is 0 (black pixels).

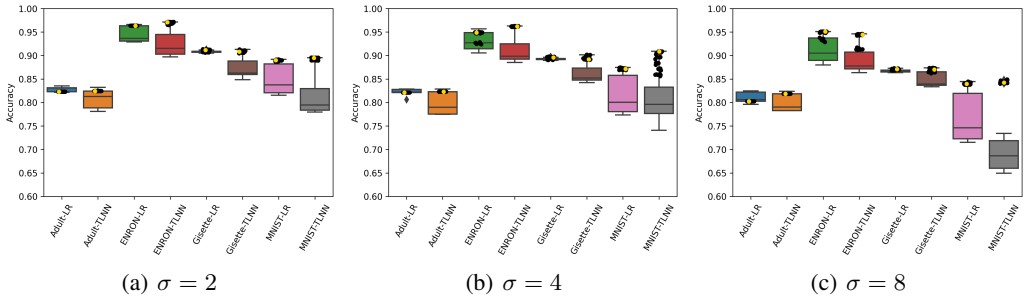

Figure 3: Ranking hyperparameter candidates across datasets. The black points correspond to the candidates with $\alpha = 0.001$ (with all permutations of $\beta_1, \beta_2$ from our searchgrid); the gold corresponds to the candidate with $\alpha = 0.001, \beta_1 = 0.9, \beta_2 = 0.999$

We observe two fundamental phenomena from this figure. First, to achieve the best accuracy, $\alpha$ and $C$ need to be tuned on a large grid spanning several orders of magnitude for each of these parameters. Second, multiple $(\alpha, C)$ pairs achieve the best accuracy and all lie on the same diagonal, validating our theory for an inverse relation between learning rate and clipping norm. As mentioned earlier, one might hypothesize that by setting the clipping norm $C$ constant and tuning $\alpha$ (corresponding to a vertical line in Figure 2) or vice versa, one could eliminate tuning a hyperparameter. However, note that not all $C$ and $\alpha$ values correspond to the lowest loss. This phenomenon is evident by noticing that not all vertical or horizontal lines on this figure have black pixels. This happens, for example, at the extremes (e.g., at the top-right corner), but also for several intermediate and standard choices (e.g., $C = 0.1$ or $0.2$). Again, the analyst has no way of knowing this a priori. We conclude that to privately tune non-adaptive optimizers, we require a large grid of hyperparameter options.

## 4.2 Tuning DP adaptive optimizers

In the interest of reducing this space of tuning we turn to *adaptive* optimizers, where we can at least reduce one dimension of this search space. These approaches automatically adapt over the learning rate $\alpha$, requiring us to tune only over the clipping norm $C$. But recall our key question: can we train models that perform competitively with the fine-tuned counterparts from DPSGD?

Adam [KB14], the canonical adaptive optimizer introduces two new hyperparameters, which are the first and second moment exponential decay parameters ($\beta_1$ and $\beta_2$). In the non-private setting, these parameters are relatively insensitive, and default values of $\alpha = 0.001, \beta_1 = 0.9$, and $\beta_2 = 0.999$ are recommended based on empirical findings, requiring no additional tuning for this hyperparameter triple. Hence before we compare DPAdam and DPSGD, we first find and establish such recommended values for this hyperparameter triple in the DP setting next, and then show that DPAdam with a small hyperparameter space performs competitively with DPSGD in Section 5.

To establish default choices of $\alpha, \beta_1$, and $\beta_2$ for DPAdam, we evaluate this private optimizer over four diverse datasets (details in Appendix B, Table 2) and two learning models including logistic regression and a neural network with one 100 neurons hidden layer (TLNN). These selected datasets include both low-dimensional data (where the number of samples greatly outnumbers the dimensionality) and high-dimensional data (where the number of samples and dimensionality are at same scale). Since we still have a large hyperparameter space to tune over, for the rest of this work, we fix a constant lot size ($L = 250$), and consider tuning over three different noise levels, $\sigma \in [2, 4, 8]$, so that we can study the effects of tuning the other hyperparameters more thoroughly. All experiments are repeated three times and averaged before reporting. Additionally, in this particular experiment since we focus on $\alpha, \beta_1$, and $\beta_2$, we also fix the clipping threshold $C = 0.5$, and $T = 2500$ iterations of training. For each dataset and model, we run DPAdam three times with hyperparameters $(\alpha, \beta_1, \beta_2)$ from the grids, $\alpha \in [0.001, 0.05, 0.01, 0.2, 0.5]$, $\beta_1, \beta_2 \in [0.8, 0.85, 0.9, 0.95, 0.99, 0.999]$.

We show that the default hyperparameter choice $(\alpha, \beta_1, \beta_2)$ of Adam in the non-private setting also works well for DPAdam. Figure 3 shows the boxplots of testing accuracies of DPAdam over the different hyperparameter choices. When $\alpha$ is 0.001 (same as in the non-private setting), all the datasets and models have final testing accuracies (marked in black) close to the best possible (and in most cases it is in fact the best) accuracy. Furthermore, we also highlight the accuracy of the

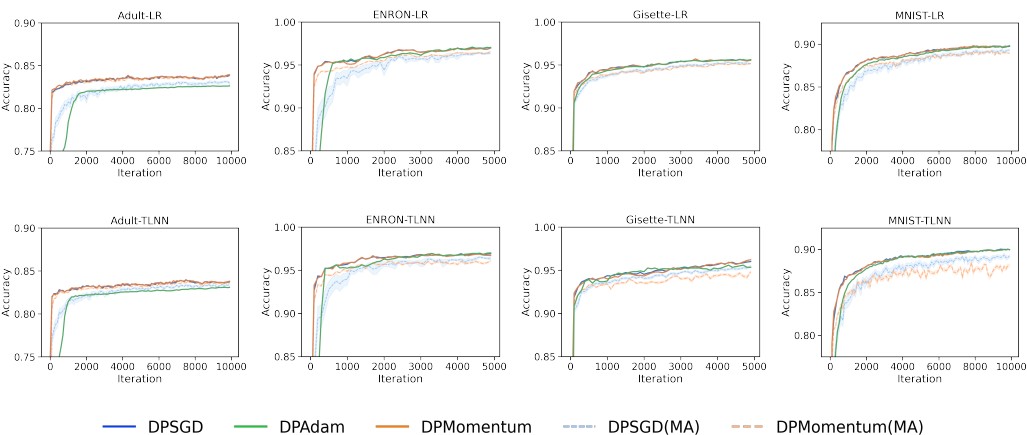

Figure 4: Comparing the testing accuracy curves of DPAdam, DPSGD and DPMomentum models across their hyperparameter tuning grids with $\sigma = 4$. The limits for y-axis are adjusted based on the dataset while maintaining a 15% range for all.

suggested default choice ($\alpha = 0.001, \beta_1 = 0.9, \beta_2 = 0.999$) using gold dots. Hence, for the ease of using DPAdam, we suggest the non-private default values for these parameters in the private setting as well and hence in all our subsequent experiments.

## 5 Advantages of tuning using DPAdam

In the non-private setting, adaptive optimizers like Adam enjoy a smaller hyperparameter tuning space than SGD. We ask two questions in this section. First, can DPAdam (with little tuning) achieve accuracy comparable to a well-tuned DPSGD? Second, what is the privacy-accuracy tradeoff one incurs when using either of the two methods we detail in Section 3 for hyperparameter selection.

To answer both questions, we compare DPAdam and DPSGD over the same set of datasets and models from the previous section. We report the accuracy of DPSGD with a range of learning rates and clipping values shown in Table 3 (Appendix C), and the testing accuracy of DPAdam with default parameter choice from Section 4.2 ($\alpha = 0.001, \beta_1 = 0.9, \beta_2 = 0.999$) and a range of clipping values $C$ in Table 3. In total, DPSGD has 40 candidates to tune over, and DPAdam has 4. This is because we have shown in Section 4.1 that DPSGD needs a wide grid to obtain the best accuracy when data distributions are unknown. Additionally, we also consider the DPMomentum optimizer. Similar to how we searched for default tuning choices for DPAdam in Section 4.2, we investigate if there exists a qualitatively good choice for the momentum hyperparameter, and unfortunately our results show that there is no such choice. We detail this process in the supplement.

In order to show the comparison from both sides of the privacy-accuracy tradeoff, we compare the three optimizers through i) the privacy cost when extracting the best accuracy from these optimizers, and ii) the accuracy one would obtain from them under the tight privacy constraints.

### 5.1 Prioritizing Accuracy

For brevity, we show experiments for $\sigma = 4$ in Figure 4, results for other values of $\sigma$ are displayed in the supplement. For each dataset and model, we train three times for each hyperparameter candidate and report the max every 100 iterations, corresponding to the dark lines for each optimizer. We note that their maxima are extremely similar. However, Table 1 shows the final privacy costs incurred by each of these max accuracy lines, and reflects our claims from Section 3.1 that using fewer hyperparameter candidates and composing privacy via MA gives a much tighter privacy guarantee.

### 5.2 Prioritizing Privacy

Additionally in Figure 4, DPSGD and DPMomentum have pastel dotted lines corresponding to their mean accuracy attained using the MA composition that provides the tightest privacy guarantees for

| Dataset | DPSGD (LT) | DPMomentum (LT) | DPAdam (MA) |
|---------|-----------|-----------------|-------------|
| Adult   | 5.01      | 5.23            | 1.91        |
| ENRON   | 30.86     | 32.31           | 12.80       |
| Gisette | 26.40     | 27.64           | 10.76       |
| MNIST   | 3.01      | 3.14            | 1.14        |

Table 1: Final $\varepsilon$ (at $\delta = 10^{-6}$) for optimizers for the LR Models (Figure 4). DPSGD and DPMomentum use LT for privacy accounting; DPAdam uses MA.

DPAdam. These pastel lines are the mean accuracies (with 95% CI) from 100 repetitions of this experiment. Since DPAdam has only 4 hyperparameter candidates, for this experiment, we sample 4 of the candidates at random for DPSGD and DPMomentum so that they all incur the same privacy cost. Since the candidate pool is significantly larger for DPSGD and DPMomentum, we additionally scrutinize the parameter grid for them and prune the learning rates that perform poorly. Our pruning process (detailed in the supplement) is quite generous, and favours minimizing the hyperparameter space of DPSGD and DPMomentum as much possible. [1] Despite the pruning advantage we see that these optimizers perform subpar than DPAdam when constrained with tight privacy requirements.

## 6   DPAdam without second moment: DPAdamWOSM

As illustrated in the previous section, adaptivity can indeed be a boon, enabling DPAdam to match the performance of tuned DPSGD while consuming roughly a third of the privacy budget as seen in Table 1. However, in addition to a decaying average of the past gradient updates, DPAdam also requires maintaining a decaying average of their second moments. In this section, we design DPAdamWOSM, a new DP optimizer that operates only using a decaying average of past gradients, as well as eliminates the need to tune the learning rate parameter. We achieve this by analyzing the convergence behavior of the second-moment decaying average in DPAdam in regimes where the scale of noise added is much higher than the scale of the clipped gradients. Setting the *effective step size* (ESS) of DPAdam to the converged constant, and removing all computations related to the second-moment updates, results in DPAdamWOSM. We empirically demonstrate that DPAdamWOSM matches the utility of DPAdam, while requiring less computation than DPAdam.

Observe that removing the second-moment updates from DPAdam reduces it to DPMomentum with one additional feature: bias-correction to the first-moment decaying average, which DPAdam does to account for its initialization at the origin. While the resultant optimizer still requires tuning the learning rate (in addition to other hyperparameters like the clipping threshold), DPAdamWOSM can be viewed as self-tuning the learning rate by fixing it to the converged effective step size in DPAdam.

### 6.1   Effective step size (ESS) in DPAdam

DPAdam produces results with a smaller variance than DPSGD due to its adaptive learning rate. To understand this phenomenon better, we look closely into the update step of DPAdam [KB14]. DPAdam being an adaptive optimizer picks per-parameter ESS as $\frac{\alpha}{\sqrt{\hat{v}_t}+\xi}$, which is the base learning rate $\alpha$ scaled by the second moment of the individual parameter gradients. We notice that when $g \to 0$, the ESS for DPAdam converges for the first moment gradient, which innately accounts for the clip bound one is training with. This may happen at later iterations, when the model is close to its minima and the gradients get close to zero.

**Theorem 3.** *The effective step size (ESS) for DPAdam with $g \to 0$ converges to $ESS^* = \frac{\alpha}{(\sigma C/L)+\xi}$.*

*Proof.* Recall that the average noisy gradient over a lot is $\tilde{g} = g + \mathcal{N}(0, \sigma^2 C^2)/L$. We now look at the effect of this noisy gradient on the effective step size (ESS) of DPAdam. As $g \to 0$, the second moment of DPAdam converges to $\frac{\sigma^2 C^2}{L^2}$. This gives us the converging value for ESS:

$$ESS^* = \frac{\alpha}{\sqrt{\hat{v}_t}+\xi} = \frac{\alpha}{\sqrt{\frac{\sigma^2 C^2}{L^2}}+\xi} = \frac{\alpha}{(\sigma C/L)+\xi}$$

---

[1]Note, pruning itself is of course unfair; the intent was to design a DP optimizer that can be used on any data distributions that we have no prior knowledge of. To do so with DPSGD one would have to consider a significantly wide range of $(\alpha, C)$ pairs to cover 'good' candidates as we illustrated in Section 4.1

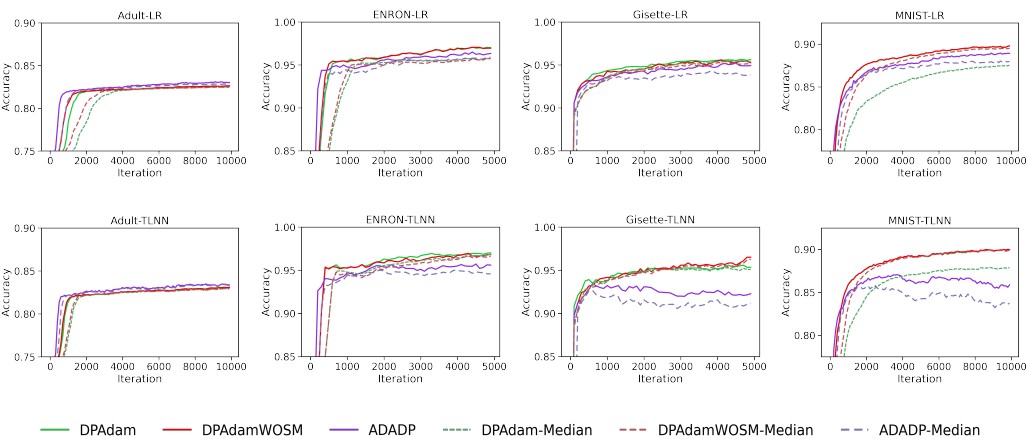

Figure 5: Comparing the testing accuracy curves of DPAdam, ADADP and DPAdamWOSM models across hyperparameter tuning grid from Table 3 with $\sigma = 4$. The limits for the y-axes are adjusted based on the dataset while maintaining a 15% range for all.

Theorem 3 gives a closed form expression that ESS converges to. We can use this value in place of $\frac{\alpha}{\sqrt{\hat{v}_t}+\xi}$ in the update step from the inception of the learning process. Since the second-moment updates (e.g., $\hat{v}_t$) are not used anymore, removing them results in our new optimizer DPAdamWOSM. We provide a pseudo-code for DPAdamWOSM in the appendix.

## 6.2 Comparing Adaptive Optimizers

We evaluate DPAdamWOSM by running it alongside DPAdam and ADADP with the same hyperparameter grid in the appendix. For brevity, we show experiments on $\sigma = 4$ and others appear in the supplement. In Figure 5, we show the maximum and median accuracy curves for all the optimizers. We display the median accuracy curves (shown in dotted), as an indicator of the quality of the entire pool of hyperparameter candidates for a given optimizer; which in this case is strictly over the choices of clip. The max lines for ADADP lies beneath DPAdam and DPAdamWOSM for all dataset except Adult. Also, the max accuracy line for DPAdamWOSM runs alongside DPAdam which means that it can perform as good as DPAdam throughout training. The median line for DPAdamWOSM also performs alongside DPAdam and in some cases is able to beat it (e.g, the median for DPAdamWOSM for MNIST-LR and MNIST-TLNN lies above the median line of DPAdam). This occurrence is seen because DPAdamWOSM uses the converged ESS from the first iteration of training.

## 7 Conclusion

In this paper, we performed a thorough investigation of honest hyperparameter selection for DP Optimizers. We compared two existing private methods, LT and MA to search for hyperparameter candidates and showed that, the former incurs a significant privacy cost but can compose a great many candidates, while the latter is helpful when the number of candidates is small. Next, we explored connections between the clipping norm and the step size hyperparameter to show an inverse relationship between them. Additionally, we compared non-adaptive and adaptive optimizers, demonstrating that the latter typically achieves more consistent performance over a variety of hyperparameter settings. This can be vital for applications where public data is scarce, resulting in difficulties when tuning hyperparameters. Finally, we brought to light that DPAdam converges to a static learning rate when the noise starts dominating the gradients. This insight allowed us to derive a novel optimizer DPAdamWOSM, a variant of DPAdam which avoids the second-moment computation and enjoys better accuracy especially at earlier iterations. Future work remains to investigate further implications of these results to provide tuning-free end-to-end private ML optimizers.

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
