# OpenReview forum: "The Role of Adaptive Optimizers for Honest Private Hyperparameter Selection"
_NeurIPS.cc/2021/Conference — NeurIPS 2021 Submitted_

### Official Review · Reviewer_iVES · 2021-07-09

**Rating:** 6
**Confidence:** 4

**Summary:**

This paper discusses the problem of private hyperparameter optimisation in machine learning. Although the discussion and results seem interesting, the privacy model is not very clear.

The response clarified this but it should be in the paper.

**Limitations And Societal Impact:**

The privacy model is unclear.

**Main Review:**

The settings seems to be this: We have a black-box optimisation algorithm $f$ that, given some hyperparamters $\theta$, it generates a model $\mu_i = f(\theta_i, D)$. The model is then evaluated on some validation set $D'$ to obtain a score $q(\mu_i, D')$, which may also be differentially privately generated. The model with the highest score is then selected. The question is, what is the overall privacy loss for $D$ if $\mu_i$ is generated with a $(\epsilon, \delta)$-DP mechanism. In this setting, we do not care about the privacy loss with respect to $D'$. The paper never makes this model perfectly clear, however.

One issue that bothered me about this paper is the fact that, in order to obtain the score, we do not need to publish the model $\mu_\theta^D$. The confusion seems to arise  from [LT19], which states that their algorithm could be used in a setting where $K$ models are published and one of them is selected. Indeed, in Sec. 3.1, line 127, the randomised mechanism samples a model $i \sim [K]$ and returns both a candidate and a score. However, the question is how much do we learn about the i-th model (and consequently D) if we only see the score on a validation dataset?
To answer this question, we must think about the DP properties of the scoring function itself on the i-th model. While it is easy to have $q$ be DP wrt D', the privacy loss wrt D is not clear. In the worst case, we can assume $\mu_i$ is fully revealed, but I find this unsatisfactory.

In any case, assuming $\mu_i$ is revealed and we do not care about privacy wrt $D'$, the results make sense and the discussion is interesting.


**Time Spent Reviewing:**

1

---

> ### Author Response · Authors · 2021-08-10
> **Response for Reviewer iVES**
>
> Thank you for your question, iVES.
>
> **Regarding the privacy model**
>
> The setting our work considers does care about privacy loss with respect to the validation set, but (as we describe briefly in lines 95-99 in our submission) we do not noise the obtained "scores" for simplicity and ease of comparison for different methods.
> Ideally, in practice, one would indeed generate the scores of each model in a differentially private fashion.
> In our work, by not noising the scores from the validation set, we implicitly treat these points as publicly available data points.
> Noising the scores will add more variance in the relative comparison of the methods considered.
> In fact, since DPSGD requires performing several more validation queries (owing to the larger hyperparameter grid for tuning) as compared to DPAdam, this would imply that DPSGD would have to either use a larger privacy budget on the validation set than DPAdam or noise the validation score more (to have the same privacy budget as DPAdam), both of which point towards the advantages of using adaptive private optimizers like DPAdam.
> Thank you for raising this issue; we will clarify this further in the paper as well.
>
> **How much do we learn about the i-th model (and consequently D) if we only see the score on a validation dataset?**
> This is indeed an interesting direction for DP learning.
> However, this is beyond the scope of this work, since both the major existing lines of work we consider in our work, MA and LT, allow releasing the final models.

---

### Official Review · Reviewer_cwxc · 2021-07-16

**Rating:** 6
**Confidence:** 3

**Summary:**

This paper presents a combination of analytical and (mostly) empirical results evaluating the task of "honest hyperparameter selection" under differential privacy; that is, accounting for the full set of tuning experiments conducted when computing the resulting privacy guarantee. The authors provide some empirical comparison between two existing composition methods for hyperparameter selections -- Moments Accountant (MA) and Liu & Talwar (LT) and give a relation between privacy of individual learners and the final learner in Theorem 1; however, their focus is mostly on showing the competitiveness of DPAdam relative to DPSGD. They conclude with a proposed adaptation to DPAdam which discards the second moments term since they observe this to converge to a constant value in practice.

**Limitations And Societal Impact:**

Yes, although I believe there could be (positive/useful) implications for DP fairness research which are not discussed; see comments above.

**Main Review:**

Overall: The paper is well-written and the experiments are generally clear. However, the ultimate intended contribution of the paper is not quite clear to me, or seems limited to providing evidence that DPAdam should be used in practice, with only the learning rate tuned, and without the second moments term. The findings are not surprising, as Adam is quite popular in practice already (although perhaps less so for DP learning, it is still available in widely available software packages like TF Privacy). Additionally, the "new" optimizer they propose mostly removes a component of Adam, and the motivation and gains from doing so are not entirely clear. I think that this could be a useful contribution to the applied DP literature, but currently it seems somewhat incremental, although there are several practical insights that are clearly expressed in the paper.

Major Comments:

* Many of the empirical results in the paper are wholly unsurprising; I am most surprised that the authors place so much emphasis on them. For example, the fact that DPAdam performs competitively with tuned DPSGD is what I would expect (and have observed in practice, as have many others) -- indeed, this is why Adam achieved such wide adoption in (non-DP) scenarios. Unless there is a strong reason to doubt this would hold in a DP context, I do not see this as an independent contribution and it only provides motivation for tuning Adam instead of SGD (which, again, seems to be the existing practice, at least outside of DP but possibly also within the DP community).

* I found the relationship between clipping bound and learning rate to be the most interesting insight from the paper, and was surprised to see less discussion and empricial investigation here. I would recommend to remove some of the other empirical results, and perhaps comment/experiment more to show what can be learned from this connection. Due to the impact that clipping is know to have on e.g. fairness and other geometric properties of DP learning, it seems that intelligently choosing the learning rate-clipping bound combination would be of great interest to many DP practitioners.

* Figure 2 is interesting, but I think the authors place a great deal of emphasis on this emprical result when it is only a single experiment with DP linear regression on what appears to be a noise-free dataset with tightly bounded features. It would be useful to see this trend across more complex models and datasets.

* The optimizer "DPAdamWOSM" is framed as a key contribution of the paper. However, it isn't clear what is gained from this new optimizer -- as the authors note, little is achieved by tuning the parameters associated with the decaying average of second moments in practice, and it is rare to tune these. So, what do we gain with respect to privacy, in practice, by simply eliminating it? The performance is effectively identical to DPAdam in Figure 5, as expected. (Of course, there are some gains in terms of computation, but the authors do not seem concerned with these.) Additionally, it seems that their method of choosing the fixed effective step size requires *running DPAdam on the data first* to set it to the converged constant. Please comment on the privacy loss of doing so.

Minor Comments:

* Please refer to the specific sections in supplementary materials when mentioning them (e.g. L132), and the specific algorithms or figures where relevant.

* I found it difficult to intuitively understand the results of Fig. 1 (right). Almost no information is given about the MA approach (apparently since the authors' findings only apply to LT), which makes it difficult to understand why the curve is shaped as it is. Furthermore, it is strange that LT is not also included in that plot (for various levels of \gamma); this would be a helpful comparison, as the authors devote little discussion to LT vs. MA in later stages and seem to choose them manually.

* L146-147 suggests that \delta_1 is much smaller than \delta_f; however, \delta_f is related to the square root of \delta_1; is this only for delta << 1?

* Please define parameters \sigma, L, T in main text.

* Fig. 2 needs more spacing from text; it is almost impossible to differentiate caption from main text.

Typos etc.

* L168: "we notice using that with the" -- please fix/clarify

L179: "blowup, and the" --> "blowup, the"

**Update**: I have read the author response. While it provides helpful clarifications and addresses some concerns, I will maintain my previous rating.

**Time Spent Reviewing:**

1.5

---

> ### Author Response · Authors · 2021-08-10
> **Response for Reviewer cwxc**
>
> **Major Comments**
>
> > 1. Many of the empirical results in the paper are wholly unsurprising; I am most surprised that the authors place so much emphasis on them. For example, the fact that DPAdam performs competitively with tuned DPSGD is what I would expect (and have observed in practice, as have many others) -- indeed, this is why Adam achieved such wide adoption in (non-DP) scenarios. Unless there is a strong reason to doubt this would hold in a DP context, I do not see this as an independent contribution and it only provides motivation for tuning Adam instead of SGD (which, again, seems to be the existing practice, at least outside of DP but possibly also within the DP community).
>
> To our knowledge, DP learning scenarios more often use DPSGD with optimal hyperparameters chosen from non-private hyperparameter tuning, as opposed to DPAdam. Furthermore, there have been works that suggest adaptive optimizers to not be suited for DP learning [1].
>
> > 2.  I found the relationship between clipping bound and learning rate to be the most interesting insight from the paper, and was surprised to see less discussion and empricial investigation here. I would recommend to remove some of the other empirical results, and perhaps comment/experiment more to show what can be learned from this connection. Due to the impact that clipping is know to have on e.g. fairness and other geometric properties of DP learning, it seems that intelligently choosing the learning rate-clipping bound combination would be of great interest to many DP practitioners.
>
> We agree that studying the effects of clipping on fairness and other geometric properties of DP learning is indeed interesting. However, it is beyond the scope of this work, as here we focus on the interplay between clipping and LR in the interest of achieving end-to-end DP learning with competitive privacy-utility tradeoffs.
>
> > 3.  Figure 2 is interesting, but I think the authors place a great deal of emphasis on this emprical result when it is only a single experiment with DP linear regression on what appears to be a noise-free dataset with tightly bounded features. It would be useful to see this trend across more complex models and datasets.
>
> We have since run the same experiments on the ENRON dataset using a logistic regression model, and we observe identical trends as seen in Figure 2 over the synthetic dataset.
> Moreover, we incorporated reviewer DFkp's suggestion of contrasting it with DPAdam training (on both synthetic and real datasets); the results from these experiments show that the lowest loss points lie along a very small LR region that covers the suggested default values, further bolstering the claim that for DPAdam the default choice of LR value is robust to a wide range (spanning several magnitudes) of the user input clipping norm.
>
> > 4. The optimizer "DPAdamWOSM" is framed as a key contribution of the paper. However, it isn't clear what is gained from this new optimizer -- as the authors note, little is achieved by tuning the parameters associated with the decaying average of second moments in practice, and it is rare to tune these. So, what do we gain with respect to privacy, in practice, by simply eliminating it? The performance is effectively identical to DPAdam in Figure 5, as expected. (Of course, there are some gains in terms of computation, but the authors do not seem concerned with these.) Additionally, it seems that their method of choosing the fixed effective step size requires running DPAdam on the data first to set it to the converged constant. Please comment on the privacy loss of doing so.
>
> There is no privacy cost involved in finding the converged learning rate. In our work, we show a closed-form expression for the effective step size in Theorem 3. This expression is used to calculate the converged learning rate from the user-defined training parameters. We then use this value as our constant learning rate for DPAdamWOSM and remove second moment computations entirely. An improvement in the training time is expected due to the removal of second moment computation. We did not measure the timing difference between the two optimizers, however, we speculate that this improvement may not be significant because of fast matrix multiplications in GPUs.
>  Nonetheless, we believe DPAdamWOSM is interesting in its own right as a simpler optimizer than DPAdam, that can attain similar performance as DPAdam while forgoing the second moment computations by leveraging the convergence property of Adam's effective learning rate in the DP learning setting.
>
> **Minor Comments**
>
> We thank you for your comments for readability and will incorporate them into the final draft.
>
> > I found it difficult to intuitively understand the results of Fig. 1 (right). Almost no information is given about the MA approach (apparently since the authors' findings only apply to LT), which makes it difficult to understand why the curve is shaped as it is. Furthermore, it is strange that LT is not also included in that plot (for various levels of \gamma); this would be a helpful comparison, as the authors devote little discussion to LT vs. MA in later stages and seem to choose them manually.
>
> Figure 1 (right) contrasts the least number of candidates one can evaluate with MA at the same privacy cost of LT. Note that Figure 1 (middle) points out that the increment in privacy cost with the number of candidates for LT is trivial. For clarity, we will add additional lines with varying $\gamma$ for LT in Figure 1 (right).
>
> > L146-147 suggests that \delta_1 is much smaller than \delta_f; however, \delta_f is related to the square root of \delta_1; is this only for delta << 1?
>
> Yes, for differential privacy $\delta_1$ is strictly $<1$, and typically we want this value to be $\ll 1/|D|$, where $D$ is the dataset.
>
>
> **References**
>
> [1] - Making the Shoe Fit. Nicolas Papernot, Steve Chien, Shuang Song, Abhradeep Thakurta, Ulfar Erlingsson. 2019.

---

> > ### Author Response · Authors · 2021-08-24
> > **Follow-up**
> >
> > We just wanted to follow-up to see if this response adequately addresses the reviewer's questions. We'd be happy to discuss any of these points more in depth.

---

### Official Review · Reviewer_akSs · 2021-07-17

**Rating:** 6
**Confidence:** 4

**Summary:**

This paper investigate and compare private hyperparameter tuning using two different methods. Further, the authors demonstrates optimal learning rate is inver proportional to optimal clipping bound. Lastly, the authors showed advantage of DP adaptive optimizer over DPSGD from the perspective of hyperparameter tuning.

**Limitations And Societal Impact:**

Limitations and suggestions are listed in the main review.

**Main Review:**

This paper presented several interesting findings, including comparing LT and MA for private hyperparameter tuning; showing relation between \sigma and C, proposing new DP optimizer. There are several concerns after reading through the paper: 1. when the authors are talking about using MA to tune hyperparameter, it is not clear what that means. MA is a method to account for the privacy cost during some randomized process. Is it referring to using grid search to find hyper parameter settings and use MA to account for the privacy cost? This part needs to be clarified. 2. The upper bound in theorem 2 looks loose. Consider the case where \sigma is 0 and C is sufficiently large such that no clipping happens at all, the bound on the right hand side becomes infinity and the convergence guarantee becomes vacant. However, in such scenario DPSGD should reduce to standard SGD with no privacy. 3. It is not clear what \alpha_opt means. What is the learning rate optimal in terms of? Is it optimal in terms of convergence bound? privacy? something else? This part needs to be clarified.

**Time Spent Reviewing:**

3

---

> ### Author Response · Authors · 2021-08-10
> **Response for Reviewer akSs**
>
> > 1. when the authors are talking about using MA to tune hyperparameter, it is not clear what that means. MA is a method to account for the privacy cost during some randomized process. Is it referring to using grid search to find hyper parameter settings and use MA to account for the privacy cost? This part needs to be clarified.
>
> Yes, we meant using grid search to find hyperparameter settings that have "good testing accuracy" and then use MA to account for the total privacy cost. For each hyperparameter candidate, the model is trained by adding necessary Gaussian noise at each iteration. We use MA as a composition mechanism to account for all these Gaussian mechanisms. We will clarify this in this final version.
>
> > 2. The upper bound in theorem 2 looks loose. Consider the case where \sigma is 0 and C is sufficiently large such that no clipping happens at all, the bound on the right hand side becomes infinity and the convergence guarantee becomes vacant. However, in such scenario DPSGD should reduce to standard SGD with no privacy.
>
> Thanks for noticing the loose bound. In our theorem and proof for Theorem 2, we assume that the gradients are being clipped and the $\sigma$ value is non-zero. If $\sigma=0$ and $C=G$, where $G$ is a notation for the upper bound on the expectation of the gradients, the convergence rate would be equal to its non-private counterpart. More precisely, in eq.1 of our proof for Theorem 2, the second term would instead be $\frac{G^2\alpha}{2}$. Moving ahead with the proof, $\alpha_{opt} = \frac{R}{G\sqrt{T}}$ and the convergence rate would be $\frac{RG}{\sqrt{T}}$ (which is similar to as in [1]). In our final version, we will specify this in text.
>
> > 3. It is not clear what \alpha_opt means. What is the learning rate optimal in terms of? Is it optimal in terms of convergence bound? privacy? something else? This part needs to be clarified.
>
> $\alpha_{opt}$ is the optimal learning rate for the convergence bound to hold. We will clarify this in the final draft.
>
> **References**
>
> [1] - Shamir, Ohad, and Tong Zhang. "Stochastic gradient descent for non-smooth optimization: Convergence results and optimal averaging schemes." International conference on machine learning. PMLR, 2013.

---

> > ### Author Response · Authors · 2021-08-24
> > **Follow-up**
> >
> > We just wanted to follow-up to see if this response adequately addresses the reviewer's questions. We'd be happy to discuss any of these points more in depth.

---

> > > ### Comment · Reviewer_akSs · 2021-08-25
> > > **Response**
> > >
> > > Thanks for your response and thanks for promising to make those changes. It has adequately addressed my concern.

---

### Official Review · Reviewer_DFkp · 2021-07-17

**Rating:** 6
**Confidence:** 3

**Summary:**

This paper looks to take a deeper dive into DP hyperparameter tuning, an often overlooked aspect of DP learning. It makes several practical claims about this problem, including comparisons of two well-known DP selection strategies and arguing for the promise of adaptive optimizers. For the latter, they provide evidence for why learning rate and clipping are difficult to tune for DP-SGD and how DPAdam can help mitigate this issue. Finally they also propose a new optimizer which matches DPAdam on performance but reduces the amount of necessary computations.

**Ethical Concerns:**

None that I have observed.

**Limitations And Societal Impact:**

Nothing to add beyond what I have mentioned previously

**Main Review:**

**Originality:** To my knowledge, the investigations/results in this paper are indeed novel and important (even if they center around existing problems/methods). I do wonder though if work on adaptive setting of the clipping threshold should be listed as an orthogonal related direction (e.g., https://arxiv.org/pdf/1905.03871.pdf, https://arxiv.org/pdf/1812.02890.pdf) – one could see these papers as also aiming to reduce the amount of tuning needed in DP training procedures, but instead focusing on $C$ instead of $\alpha$.

**Quality:** I find the results solid for the most part but I have a few comments/questions here that I hope the authors can speak to.

(1) Was the choice to fix $\alpha$ in Sec 5 taken after inspecting the results in Sec 4.2? If so, is it actually fair to use the reduced search space for DPAdam in Sec 5 if the this reduction was based on inspecting additional runs on the same datasets earlier in Sec 4.2? It seems it would be stronger/more generalizable to see if these default choices carry over to do well on unseen tasks.

(2) I think there are a few places where more extensive experiments might help better support the author's claims. These generally relate to getting a more precise sense of DPAdam's robustness over choices of $\alpha$.

* (a) A version of Figure 2 for DPAdam would be appreciated to compare with the existing one for DPSGD. Additionally, maybe similar analyses could also be repeated on more realistic datasets, especially if the aim is to conclude more generally that “to privately tune non-adaptive optimizers, we require a large grid.”

* (b) While I buy that DPAdam is very likely easier to tune, I still question the generalizability of the stronger takeaway that one is recommended to fix the same default learning rate for DPAdam on new tasks. In practice, people do see non-trivial gains from tuning non-private Adam’s learning rate and indeed, it does seem like accuracy is hampered in the case of Adult by using the default. Perhaps experiments covering more graded reductions of the search space/privacy budgets could help clarify best practices for DPAdam.

* (c) In Section 5, I’d be curious to see for completeness how DPAdam does when using the full search space (of DPSGD).

(3) I wonder if the authors could speak more about the significance or practical benefit DPAdamWOSM offers. Is the claim is based on computational efficiency rather than accuracy/privacy (which look comparable to DPAdam)? Is so, what do the actual savings look like here?

**Clarity:** As a whole, the paper reads well.  However, I did notice a few misc. things that I wanted to note

* In Sec 1.1, it is said that [CMS11, CV13, KGGW15] “work either in restricted settings...or under relaxations of differential privacy.” Given that $(\epsilon, \delta)$-DP, which is studied in this paper, itself is considered a relaxation of “pure” $(\epsilon)$-DP, the current wording might be confusing if it aims to communicate how these previous works differ.

* Were the quantities $\sigma, L, T$ defined prior to their usage in line 150? I feel that presentation would be helped if the full set of hyperparameters were explained in the main (currently left to appendix A) or at least mentioned with a pointer to the specific Appendix for more details. Similarly, ADADP is introduced without much context in Figure 5/Sec 6.2,

* Why is it said that $\delta_2$ has little effect on $\delta_f$ in line 138? If we set it arbitrarily small (such as to the indicated 10^-20), doesn’t this end up blowing up $\delta_f$ by a factor of 20 (via the dependence of $\Upsilon$ on $\delta_2$)?

* It’s kind of difficult to actually see the “black” squares in Figure 2. Also wondering if these losses here were averaged over multiple runs?

**Significance:** Overall, I think the contributions in this paper are interesting and tackle a very well-motivated problem. The privacy costs incurred by hyperparameter tuning are indeed an underemphasized aspect of “honest” DP model training. It seems natural then to study adaptive optimizers, which likely have an even larger role to play than in non-private settings. Here, one does not merely aim to conveniently save time/resources in the DP setting, but also the amount of necessary noise added to training procedures (assuming limited privacy budget). In this context, I find the major claims to all be of practical value towards bridging this gap to "honest" DP selection. However I do think some of them can be more extensively evaluated/supported (as discussed above).


**Time Spent Reviewing:**

6

---

> ### Author Response · Authors · 2021-08-10
> **Response for Reviewer DFkp**
>
> **Quality**
> >1. Is the choice to fix $\alpha$ in Sec 5 taken after inspecting the results in Sec 4.2? If so, is it actually fair to use the reduced search space >for DPAdam in Sec 5 if the this reduction was based on inspecting additional runs on the same datasets earlier in Sec 4.2? It seems it >would be stronger/more generalizable to see if these default choices carry over to do well on unseen tasks.
>
> Thank you for that observation.
> We agree that if we were in fact choosing new default hyperparameters based on the experiments from 4.2 it would be unfair, and we would end up selecting hyperparameters that could have been overfitted to these datasets.
> However, the tuning in Section 4.2 was to validate whether the default choices of non-private Adam perform competitively for DPAdam, and through our experiments on various datasets and tasks, we show that this is true.
>
> >2. a) A version of Figure 2 for DPAdam would be appreciated to compare with the existing one for DPSGD. Additionally, maybe similar analyses could also be repeated on more realistic datasets.
>
> Thanks for the suggestion. For a better understanding of the simulation experiment, we have rerun the simulation experiment on the ENRON dataset using a Logistic Regression model. We have generated two separate plots for DPSGD and DPAdam, which we are not able to post in the reply for regulatory reasons.
> Instead, we will add our inferences from these plots here. The DPSGD graph shows a similar trend to Figure 2 in the paper backing up the inverse relation between LR and C.
> On the other hand, DPAdam shows a different trend. The black pixels (lowest loss) for DPAdam lie along a straight line at its default LR value $\alpha = 0.001$. This shows that the competitive performance of DPAdam with its default LR value is robust to a wide range (spanning several magnitudes) of the user input clipping norm.
> We will add the new plots to the final version of our paper.
> > 2. b) While I buy that DPAdam is very likely easier to tune, I still question the generalizability of the stronger takeaway that one is recommended to fix the same default learning rate for DPAdam on new tasks. Perhaps experiments covering more graded reductions of the search space/privacy budgets could help clarify best practices for DPAdam.
>
> We agree that additional tuning of the suggested default learning rate for DPAdam over the task at hand can only further help utility.
> Our main goal with this work was to provide guidance for data analysts who wish to deploy models with end-to-end DP guarantees.
> In particular, in several scenarios they may achieve better privacy-utility tradeoffs by performing minimal tuning with DPAdam, than expending exorbitant compute and higher privacy budget (both of which arise from the significantly larger grid of hyperparameters) for DPSGD.
>
> > 2. c) In Section 5, I’d be curious to see for completeness how DPAdam does when using the full search space (of DPSGD).
>
> We now have additional graphs that address this point, and we will add them in the final version of the paper. The summary of our results is that while using the full grid as DPSGD, for the most part, does not improve performance on DPAdam as compared to the constrained grid (it does improve accuracy by $\approx 1$% for the Adult dataset), we find that DPAdam is indeed less sensitive than DPSGD to the choices of learning rate.
> We note that the full grid choices for LR span 4 orders of magnitude, and yet we find that the default choice of LR performs competitively with the best candidates from the full grid.
>
> > 3. I wonder if the authors could speak more about the significance or practical benefit DPAdamWOSM offers. Is the claim is based on computational efficiency rather than accuracy/privacy (which look comparable to DPAdam)? Is so, what do the actual savings look like here?
>
> DPAdamWOSM can be looked at as DPAdam but without the second moment computation and setting learning rate $\alpha$ to the converged constant value of ESS from Theorem 3 for the complete training procedure. We have observed a minor improvement in testing accuracy at the start of training for DPAdamWOSM. An example of this can be seen when the median line of DPAdamWOSM lies above that of DPAdam for MNIST TLNN in Figure 5.
> An improvement in the training time is expected due to the removal of second moment computation.
> However, this improvement is unlikely to be significant because of fast matrix multiplications on the GPU.
> Nonetheless, we believe DPAdamWOSM is interesting in its own right as a simpler optimizer than DPAdam, that can attain similar performance as DPAdam while forgoing the second moment computations by leveraging the convergence property of Adam's effective learning rate in the DP learning setting.
>
> **Clarity**
>
> > In Sec 1.1, it is said that [CMS11, CV13, KGGW15] “work either in restricted settings...or under relaxations of differential privacy.” Given that $(\epsilon,\delta)$-DP, which is studied in this paper, itself is considered a relaxation of “pure” ($\epsilon$)-DP, the current wording might be confusing if it aims to communicate how these previous works differ.
>
> We will rephrase it to ``under relaxation of approximate differential privacy" to avoid the ambiguity w.r.t. pure DP.
>
> > Were the quantities $\sigma$, L, T defined prior to their usage in line 150? I feel that presentation would be helped if the full set of hyperparameters were explained in the main (currently left to appendix A) or at least mentioned with a pointer to the specific Appendix for more details. Similarly, ADADP is introduced without much context in Figure 5/Sec 6.2,
>
> Due to space constraints, we had to move parts of our preliminary to the Appendix. We will add references to the Appendix in the main text for better readability.
>
> >Why is it said that δ2 has little effect on δf in line 138? If we set it arbitrarily small (such as to the indicated 10^-20), doesn’t this end up blowing up δf by a factor of 20 (via the dependence of $\Upsilon$ on δ2)?
>
> Indeed as you correctly point out, setting the $\delta_2$  parameter has an effect on $\delta_f$. However, $\delta_f$ is usually set to a user-defined value ($\ll 1/|D|$). In our experiments, we simulate this by setting the $\delta_f$ constant and searching for the minimum $\delta_1$ using a ternary search algorithm. We then compute the corresponding $\epsilon_1$ from Moments Accountant for the $\delta_1$ calculated from the search algorithm.
> We have a new table that shows how the final epsilons change at varying values of $\delta_2$ for different dataset sizes. The table shows that for a $5k$ sized dataset, the final epsilon varies from $32.56$ to $33.63$ for $\delta_2$ of $10^{-10}$ and $10^{-30}$ respectively. Similarly, for a $100k$ dataset, these values range from $1.73$ to $1.78$ respectively.  We note that the $\epsilon_1$ doesn't change much with the starting choice of $\delta_2$, and that this effect is best observed for large datasets as seen from the epsilon ranges for $5k$ and $100k$ sized datasets.
>
> > It’s kind of difficult to actually see the “black” squares in Figure 2. Also wondering if these losses here were averaged over multiple runs?
>
> Thank you for the suggestion. In our next version of the paper, we will incorporate a different color scheme which will enable better readability. We had run the experiment in Figure 2 once. However, we repeated the experiment 3 times and got similar results as Figure 2.

---

> > ### Author Response · Authors · 2021-08-24
> > **Follow-up**
> >
> > We just wanted to follow-up to see if this response adequately addresses the reviewer's questions. We'd be happy to discuss any of these points more in depth, or other details of our work.

---

### Decision · Program_Chairs · 2021-09-27

**Decision:**

Reject

**Comment:**

This paper studies the problem of hyper-parameter selection in differential private learning, making a number of observations on the relation between learning rate and clipping parameters, and showing that adaptive optimizers like DP-Adam can outperform standard ones that are more commonly used in practice (eg, DP-SGD). One of the merits of the paper is to tackle an important but under-studied problem (hyper-parameter optimization under DP) from a practical standpoint, accounting for the privacy loss of tuning and showing that this leads to different trade-offs in the choice of optimizers.

The reviews were all somewhat positive but none of the reviewers expressed clear support for the paper acceptance. The author response helped to clarify a few aspects, but was not sufficient to get stronger support. Remaining concerns/doubts include:
- Insufficient evidence that DP-Adam is currently overlooked (the authors only cite one paper), which is crucial to assessing the significance of the DP-Adam findings
- The results seem to rely on the fact that default parameters for adaptive optimizers are good enough, which may not always be the case in practice.
- The new experiments replicating Figure 2 were found to be interesting but it is difficult to assess their completely without being able to see them in full.

Eventually, the decision was to reject the paper. However, the work is seen as promising so the authors are strongly encouraged to improve their work based on the feedback from the reviewers.